# ZERO-SHOT QUANTIZATION FOR OBJECT DETECTION

## ABSTRACT

Zero-shot quantization (ZSQ) has achieved remarkable success in classification tasks by leveraging synthetic data for network quantization without accessing the original training data. However, when applied to object detection networks, current ZSQ methods fail due to the inherent complexity of the task, which encompasses both localization and classification challenges. On the one hand, the precise location and size of objects within the samples for object detection remain unknown and elusive in zero-shot scenarios, precluding artificial reconstruction without ground-truth information. On the other hand, object detection datasets typically exhibit category imbalance, and random category sampling methods designed for classification tasks cannot capture this information. To tackle these challenges, we propose a novel ZSQ framework specifically tailored for object detection. The proposed framework comprises two key steps: First, we employ a novel bounding box and category sampling strategy in the calibration set generation process to infer the original training data from a pre-trained detection network and reconstruct the location, size and category distribution of objects within the data without any prior knowledge. Second, we incorporate feature-level alignment into the Quantization Aware Training (QAT) process, further amplifying its efficacy through the integration of feature-level distillation. Extensive experiments conducted on the MS-COCO and Pascal VOC datasets demonstrate the efficiency and state-of-the-art performance of our method in low-bit-width quantization. For instance, when quantizing YOLOv5-m to 5-bit, we achieve a 4.2% improvement in the mAP metric, utilizing only about 1/60 of the calibration data required by commonly used LSQ trained with full trainset.

## 1 INTRODUCTION

Object detection neural networks play a pivotal role in a wide array of computer vision applications, spanning from autonomous driving to surveillance systems (Mao et al., 2023; Balasubramaniam & Pasricha, 2022; Oguine et al., 2022; Mishra & Saroha, 2016). As the demand grows for deploying deep neural networks on resource-constrained devices, quantization has emerged as a critical technique to reduce network size and computational complexity while maintaining performance (Chen et al., 2019; Deng et al., 2020; Han et al., 2015; Wang et al., 2022). However, traditional quantization methods often necessitate access to the original training data, posing challenges due to privacy concerns or the impracticality of storing and transferring large datasets (Krishnamoorthi, 2018a; Nagel et al., 2021a). In this context, Zero-shot Quantization (ZSQ) (Cai et al., 2020; Nagel et al., 2019a; Yvinec et al., 2023; Xu et al., 2020; Liu et al., 2021) presents a promising approach to quantize neural network without the reliance on real training data, which mainly leverage synthetic data inversed from network with randomly sampled label. Most current research on ZSQ is limited to classification tasks and leverages synthetic data to fine-tune the quantized network in a distillation manner, offering a pathway to privacy-preserving quantization (Cai et al., 2020; Xu et al., 2020; Liu et al., 2021).

While ZSQ has achieved remarkable success in classification tasks, the extension of zero-shot techniques to object detection faces unique challenges due to the inherent complexity of the object detection task that encompasses both localization and classification subtasks. First, the existing data synthesizing method of zero-shot quantization cannot be extended to object detection. Classification networks require only a randomly sampled category id label for data synthesizing (Choi et al., 2021; Zhong et al., 2022; Qian et al., 2023a;b; Chen et al., 2024). Unlike classification networks, the location and size of objects within the samples for object detection remain unknown and elusive in zero-shot scenarios, precluding artificial reconstruction without ground-truth information. Random

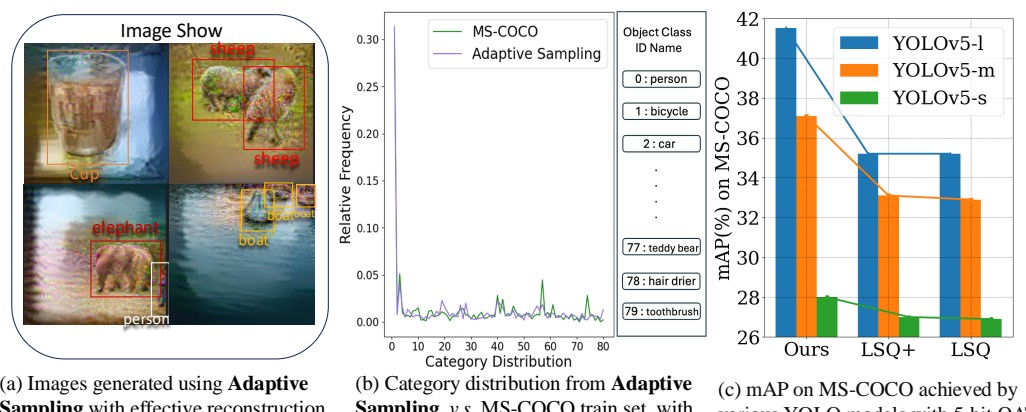

(a) Images generated using **Adaptive Sampling** with effective reconstruction of object location information.

(b) Category distribution from **Adaptive Sampling** *v.s.* MS-COCO train set, with object category names listed on the right.

(c) mAP on MS-COCO achieved by various YOLO models with 5-bit QAT.

Figure 1: (a) Images generated by **Adaptive Sampling** on a YOLOv5 detector pre-trained on MS-COCO. (b) **Adaptive Sampling** can generate a category distribution frequency similar to MS-COCO in a data-free setting. (c) We achieve SOTA under all settings where LSQ and LSQ+ use 120K real data for training, while ours uses the 2K calibration set for training.

sampling the location and size tends to result in bad plausibility of the relative positions and sizes of the objects. Chawla et al. (2021); Chen et al.; Wang et al. (2024) focus on synthetic detection sample and take ground-truth as the label for generation, which can accurately reconstruct the location and size of the detection samples, but ground-truths are not allowed in the zero-shot settings and internal information of the network such as batch normalization statistics is not considered. On the other hand, object detection datasets typically exhibit category imbalance, and random category sampling methods designed for classification tasks like category-balanced ImageNet and CIFAR-10/CIFAR-100 cannot capture the category distribution of objects. Second, the fine-tuning strategy for zero-shot quantized detection network with synthetic calibration data has not been studied. The currently used logits alignments designed for classification networks might not be sufficient in more complex object detection networks, which makes it hard to make full use of limited synthetic data efficiently.

To address these challenges, we propose a novel ZSQ framework for quantizing object detection networks. We first propose a novel bounding box and category sampling strategy to synthesize a calibration set from a pre-trained detection network, which reconstructs the location, size and category distribution of objects within the data without any prior knowledge. Then, we integrate prediction-matching distillation and feature-level distillation into the Quantization Aware Training (QAT) process to further amplify the efficacy of quantized detection network finetuning. For the first time, we demonstrate synthetic calibration set can be applied to object detection network quantization, particularly in privacy-sensitive scenarios. Encouragingly, compared with the time and resource-intensive quantization-aware training (QAT) method LSQ based on the full training set, it is shown that our synthetic calibration set, merely 1/60 the size of the original training set, can yield comparable results with our framework.

Specifically, Our contributions are threefold and are visually illustrated in Fig. 1:

1. **Location reconstruction.** To reconstruction of position and size of the objects in synthetic samples for zero-shot quantization, the proposed bounding box sampling method tailored for object detection exhibit plausibility of the relative positions and sizes of the objects, showcasing that potential object localization information can be obtained from the network.

2. **Category sampling.** To inverse the category-imbalance object detection model, we employ a relabel strategy for category sampling strategy in the calibration set generation process and reconstruct the category distribution of objects within the synthetic samples without any prior knowledge.

3. **Advanced performance.** We integrate knowledge distillation into the quantized object detection network fine-tuning process for better knowledge transfer. Extensive experiments conducted on the MS-COCO and Pascal VOC datasets demonstrate the state-of-the-art performance of our method in zero-shot quantization. For instance, when quantizing

YOLOv5-m to 5-bit, we improve the mAP metric by 4.2% compared to LSQ trained with full real data.

## 2 RELATED WORKS

This section provides a brief overview of the studies relevant to our work, focusing on data-driven quantization and zero-shot quantization.

**Data-driven Quantization**  Post-training quantization (PTQ) and quantization-aware training (QAT) (Krishnamoorthi, 2018b; Nagel et al., 2021b) are the most commonly employed quantization methods. PTQ methods typically utilize a small calibration set, often a subset of the training data, to optimize or fine-tune quantized networks (Finkelstein et al., 2023; Frantar & Alistarh, 2022). For instance, AdaRound (Nagel et al., 2020) introduced a layer-wise adaptive rounding strategy, challenging the quantizers of rounding to the nearest value. Additionally, BRECQ (Li et al., 2021) implemented block-wise and stage-wise reconstruction techniques, striking a balance between layer-wise and network-wise approaches. QDrop (Wei et al., 2022) innovatively proposed randomly dropping activation quantization during block construction to achieve more uniformly optimized weights. Despite their simplicity and minimal data requirements, PTQ methods often face challenges related to local optima due to the limited calibration set available for fine-tuning. On the other hand, most QAT approaches leverage the entire training dataset to quantize networks during the training process (Jung et al., 2019). PACT (Choi et al., 2018) introduced a parameterized clipping activation technique to optimize the activation clipping parameter dynamically during training, thereby determining the appropriate quantization scale. LSQ (Esser et al., 2019) proposed estimating the loss gradient of the quantizer's step size and learning the scale parameters alongside other network parameters. LSQ+ (Bhalgat et al., 2020), an extension of the LSQ method, introduced a versatile asymmetric quantization scheme with trainable scale and offset parameters capable of adapting to negative activations. Both QAT and PTQ methods rely on training data for quantization, rendering them impractical when faced with privacy or confidentiality constraints on the training data.

**Zero-shot Quantization**  Zero-Shot Quantization (ZSQ) is a valuable approach that eliminates access to real training data during the quantization process. Presently, most ZSQ research is confined to classification tasks. Data-free quantization (DFQ) represents a subset of ZSQ methods that enable quantization without relying on any data. For instance, DFQ (Nagel et al., 2019b) introduced a scale-equivariance property of activation functions to normalize the weight ranges across the network. SQuant (Guo et al., 2022) developed an efficient data-free quantization algorithm that does not involve back-propagation, utilizing diagonal Hessian approximation. However, due to the absence of data, DFQ methods may not be suitable for low-bit-width configurations. For example, in the case of 4-bit MobileNet-V1 on ImageNet, SQuant achieved only 10.32% top-1 accuracy. Another branch of ZSQ methods leverages synthetic data (Chen et al., 2023; Li et al., 2023) generated by the full-precision network. GDFQ (Shoukai et al., 2020) introduced a knowledge-matching generator to synthesize label-oriented data using cross-entropy loss and batch normalization statistics (BNS) alignment. TexQ (Chen et al., 2024) emphasized the detailed texture feature distribution in real samples and devised texture calibration for data generation. However, the algorithms designed for classification tasks may not be directly applicable to detection tasks, as they cannot effectively utilize the output of the detection head. From the perspective of synthetic samples with detection networks, Chawla et al. (2021); Chen et al.; Wang et al. (2024) presented relevant methods to synthesize data with ground-truth or small amounts of real data for distillation or network training. However, both ground-truth and real samples should be prohibited in the zero-shot settings and they were not aimed at zero-shot quantization tasks and therefore lacked consideration of the internal information of the model.

## 3 METHODOLOGY

In this section, we provide an overview of the proposed framework, as illustrated in Fig. 2. Our framework consists of two stages: generation of a condensed calibration set and quantization-aware training (QAT).

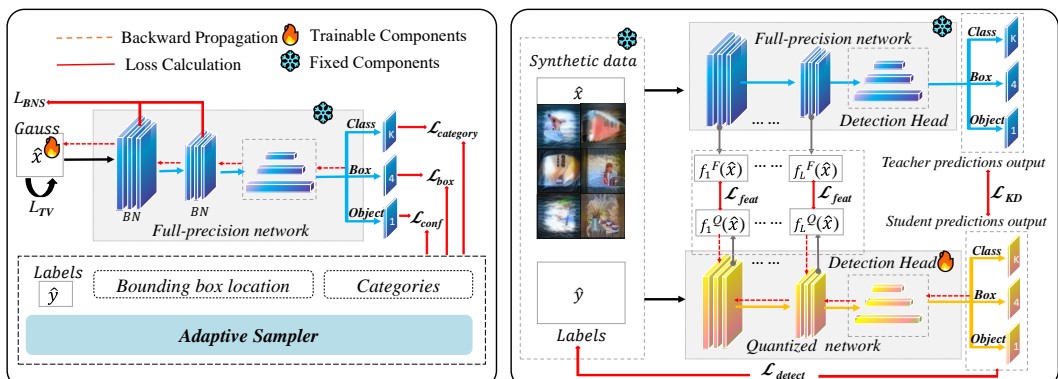

Figure 2: An overview of the proposed framework.

## 3.1 PRELIMINARIES

**Quantizer.** Following LSQ (Esser et al., 2019), we adopt per-tensor symmetric quantization on both weights and activations. Given a floating-point tensor $w_{fp}$ (weights or activations), step size $s$ and quantization bit width $b$, the quantized representation of the data $\hat{w_{fp}}$ can be defined as:

$$w_{int} = clip(\lfloor w_{fp}/s \rceil, -2^{b-1}, 2^{b-1} - 1),$$  (1)

$$\hat{w_{fp}} = w_{int} \times s.$$  (2)

Here, $w_{int}$ denotes the quantized integer representation of the data, $\lfloor input \rceil$ rounds the input to its nearest integer. We conduct quantization on a 32-bit floating-point full-precision pre-trained YOLO network. During optimization finetuning, We follow LSQ (Esser et al., 2019) to update the weight parameters and step size.

**Calibration Set Generation.** The calibration set utilized in model quantization needs to reflect the model's inherent distribution. Data-free quantization seeks to generate a synthetic calibration set that matches the model's distribution (Cai et al., 2020). The synthetic calibration set can be derived through noise optimization (Cai et al., 2020; Zhong et al., 2022; Zhang et al., 2021), which is usually instantiated by distribution approximation (Cai et al., 2020; Xu et al., 2020). Existing data-free methods in object detection typically require generating a calibration set the same size as the training set (120k images for MS-COCO) (Chawla et al., 2021). In contrast, we only generate a small amount of calibration set to extract features of the data.

Given a batch of N inputs $x \in R^{N \times 3 \times H \times W}$, where each pixel is initialized from random Gaussian noise $x_{i,c,h,w} \sim \mathcal{N}(0,1)$, and a pre-trained full-precision detection network $\phi(\theta)$, synthetic calibration set are obtained through optimizing the inputs to match the batch normalization statistics (BNS) (Yin et al., 2020):

$$\min_x \mathcal{L}_{BNS}(x) = \sum_{l=1}^{L} (||\mu^l(\theta, x) - \mu^l(\theta)||_2 + ||\sigma^l(\theta, x) - \sigma^l(\theta)||_2),$$  (3)

where $\mu^l(\theta)/\sigma^l(\theta)$ are mean/variance parameters stored in the $l$-th BN layer of $\phi(\theta)$ and $\mu^l(\theta, x)/\sigma^l(\theta, x)$ are mean/variance parameters calculated on inputs using $\phi(\theta)$. It enforces feature similarities at all levels by minimizing the distance between the feature map statistics for the synthesized image $x$ and the real image $\hat{x}$.

Besides the BNS alignment objective function, a network training loss is also utilized to optimize the sampled inputs. In classification networks, the standard form is $L_{classify}(\phi(x), c)$, where the target label c is an integer and can be generated through random sampling. In object detection tasks, however, the label information is more intricate, often comprising the object's position and size,

which can be formulated as: $L_{detect}(\phi(x), y)$. This consists of three components: a box category loss $\mathcal{L}_{category}$, a box dimension loss $\mathcal{L}_{box}$, and a grid location loss $\mathcal{L}_{conf}$. The ground truth target $y \in \mathcal{R}^{N \times 6}$ includes the batch index (y[:, 0]) $i$, the category of the bounding box (y[:, 1]) $c$, and the coordinates of the bounding box (y[:, 2:6]) $x,y,w,h$. More specifically, a prior term consisting of the total variance and $l_2$ norm of the input image is always involved in the final loss function to steer images away from unrealistic images (Mahendran & Vedaldi, 2015):

$$\min_x \mathcal{L}_{prior}(x) = \alpha_{TV} \mathcal{L}_{TV}(x) + \alpha_{l_2} \|x\|_2^2, \tag{4}$$

where $\mathcal{L}_{TV}$ promotes similarity between adjacent pixels by minimizing their Frobenius norm, consequently enhancing the smoothness of the synthetic image, $\alpha_{TV}$ and $\alpha_{l_2}$ are hyper-parameters balancing the importance of two terms. Finally, we can regard our framework as a regularized minimization problem and optimize the following function:

$$\min_x \alpha_{BNS}\mathcal{L}_{BNS}(x) + \alpha_{detect}\mathcal{L}_{detect}(\phi(x), \mathbf{y}) + \mathcal{L}_{prior}(x). \tag{5}$$

## 3.2 ADAPTIVE SAMPLING

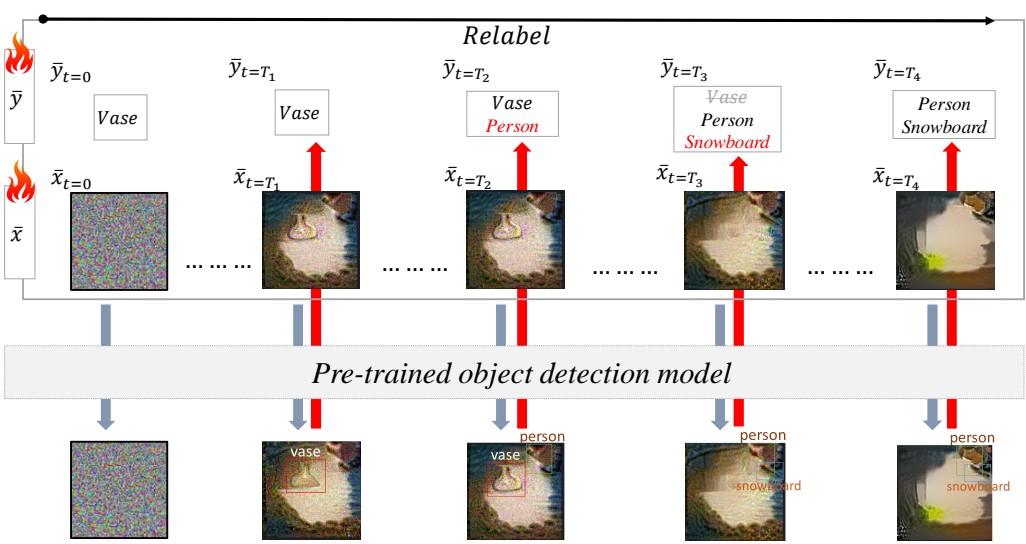

Figure 3: An overview of **Relabel** process.

In this section, we propose an adaptive sampling strategy to sample bounding box coordinates and categories needed for generating the calibration set in the Section 3.1. The proposed adaptive sampling strategy only requires a pre-trained network and does not rely on additional information (e.g. meta-data, feature activation) or additional networks(e.g. pre-trained generative networks).

Unlike classification networks, the inversion of object detection networks presents several challenges. First, the location and size of objects within the network are unknown. In contrast to classification networks, where classification labels can be randomly sampled, the labels for object detection networks include both object categories and bounding boxes. Therefore, random sampling faces challenges related to the plausibility of the relative positions and sizes of the objects. As shown in Section D, the performance of random sampling using given multi-object labels is significantly worse than that of our adaptive sampling method. Besides, object categories distribution is also unknown. Current method typically uses ground truth to guide model inversion, which is not truly zero-shot. In the field of object detection, the DIODE (Chawla et al., 2021) method introduced the use of positive samples, but it lacks a mechanism for adjusting negative samples and has not been applied to the field of zero-shot quantization.

Table 1: Bounding box sampling details: we start by sampling one object $\mathbf{Y}$ for each image, where $\mathbf{C}$ represents the number of categories. We assume that the relative width and height of the image are both 1. $W\_min$ and $H\_min$ are set to 0.2, while $W\_max$ and $H\_max$ are set to 0.8. $\mathcal{U}$ denotes uniform distribution.

| Variable | Sampling Distribution | Description |
|----------|----------------------|-------------|
| $\mathbf{Y}[i,0]$ | - | Batch index |
| $\mathbf{Y}[i,1]$ | $\mathcal{U}(0, C)$ | Category |
| $\mathbf{Y}[i,2]$ | $\mathcal{U}(W/2, 1 - W/2)$ | Bouding box x-center |
| $\mathbf{Y}[i,3]$ | $\mathcal{U}(H/2, 1 - H/2)$ | Bouding box y-center |
| $\mathbf{Y}[i,4]$ | $\mathcal{U}(W\_min, W\_max)$ | Bouding box width |
| $\mathbf{Y}[i,5]$ | $\mathcal{U}(H\_min, H\_max)$ | Bouding box height |

Motivated by (Yin et al., 2024), which integrates soft labels into the data recovery process, making synthetic data and labels more aligned, we propose a two-stage label sampling method, outlined as follows:

1. **Relabel.** The first stage is responsible for generating labels through relabeling, as illustrated in Fig. 3. We start by sampling one object $Y \in \mathcal{R}^{K \times 6}$ for each image $x \in \mathcal{R}^{3 \times H \times W}$ in the batch, details are included in Table 1. While optimizing the input toward the generated targets using Eq. 5, we use a pre-trained teacher detection network to relabel the synthetic images according to Alg. 1 every 100 iterations. We aggregate labels with high confidence in the label space of the teacher's dataset and remove labels with low confidence, ensuring at least one label in each image.

2. **Synthetic.** In the second stage, we fix the generated labels and optimize the input towards the targets using Eq. 5. The main difference from the calibration set generation process is that the labels we use are obtained through relabeling samples, rather than real labels.

With this sampling strategy, we eliminate the need for real detection labels. As presented in Fig. 1, this approach can produce bounding box categories that closely resemble the actual distribution, while also reconstructing objects' relative positions, sizes, and counts. This capability supports downstream tasks, such as quantization-aware training, in a data-free setting.

---

**Algorithm 1** Adaptive Sampling Algorithm

---

**Input:** existing image and labels $\{image, targets\}$, pre-trained detection network $teacher$, filtering threshold: confidence $conf\_thresh$, iou $iou\_thresh$

1. $new\_targets = teacher(image).predictions[conf > conf\_thresh]$
2. $ious = \text{IOU}(new\_targets, targets)$
# Add targets that do not overlap with the existing targets.
3. $add\_targets = new\_targets[(\max(ious, dim = 1) < iou\_thresh).bool()]$
# Remove targets from the existing list that are not detected by the pre-trained detection network.
4. $minus\_targets = (\max(ious, dim = 0) < iou\_thresh).bool()$
5. $targets = targets[\sim minus\_targets]$
6. $targets = \text{cat}([targets, add\_targets], dim = 0)$

---

### 3.3 EFFICIENT FINE-TUNING WITH DISTILLATION

In this section, we propose to reduce the knowledge discrepancy between full-precision pre-trained network (teacher) and quantized network (student) through knowledge distillation.

Knowledge distillation (Hinton et al., 2015) is a commonly used method for knowledge transfer. Previous works (Ding et al., 2023; Li et al., 2023) have applied it to classification tasks with quantized CNNs and LLMs for better performance. This indicates that aligning quantized networks in the feature dimension is beneficial for maintaining network performance. The backbone and prediction head of a full-precision pre-trained YOLO network contains much of the statistical information from real training data (Yin et al., 2020), which cannot be fully explored by object detection loss. Therefore,

we propose using feature-level distillation to match intermediate features and prediction-matching distillation to align the predictions of the quantized network and pre-trained network.

**Prediction-matching Distillation**    As proposed in Section 3.1, our synthetic calibration set $\{(\hat{x}^i, \hat{y}^i)\}_{i=1}^N$ is the result of the network backpropagating through pre-defined labels, directly aligning predictions of the quantized network with the targets would lead to severe over-fitting issues. Therefore, we introduce Kullback–Leibler (KL) divergence loss (Kullback & Leibler, 1951) between the predictions of quantized network and full-precision network as soft labels and object detection loss between the predictions of quantized network and targets as hard labels to recover the performance of the quantized network, which is represented as:

$$\min_{\theta'} \mathcal{L}_{KD} = \beta_{KL} \frac{\tau^2}{N} \sum_{i=1}^N KL(z^F(\hat{x}_i; \theta), z^Q(\hat{x}_i; \theta')) + \beta_{detect} \frac{1}{N} \sum_{i=1}^N \mathcal{L}_{detect}(\phi'(\hat{x}_i), \hat{y}_i), \quad (6)$$

where $\{\hat{x}_i\}_{i=1}^N$ is a batch of the calibration set images, $z^F(\hat{x}_i; \theta)/z^Q(\hat{x}_i; \theta')$ are output predictions from full-precision / quantized network and $\tau$ is the distilling temperature. We denote parameters of full-precision / quantized network as $\theta/\theta'$, $\beta_{KL}$ and $\beta_{detect}$ are hyper-parameters to balance the two terms.

**Feature-level Distillation**    We extend the knowledge transfer manner to the feature level and introduce a feature distillation method to match intermediate features from teacher and student explicitly. We find the benefits are two-fold: On the one hand, it accelerates the network convergence. Typical LSQ methods train the network's weight parameters and quantization step size with a whole set of real images. In contrast, as we will demonstrate in Section 4.1, through finer knowledge transfer, we enable the quantized student to train on a synthetic calibration set of merely 1/60 of the original size and boost the convergence speed by about $16\times$. On the other hand, QAT training at ultra-low bit width always leads to rapid error accumulation. Feature distillation ensures the similarity of features extracted by the teacher and student, thereby minimizing error accumulation during the training process.

In the quantization-aware training stage, given a batch of synthetic image $\{\hat{x}_i\}_{i=1}^N$, we impose the mean squared error constraints between the feature maps from teachers and students. With $L$ being the number of distilling network layers, the feature distillation loss $\mathcal{L}_{feat}$ can be expressed as:

$$\min_{\theta'} \mathcal{L}_{feat} = \frac{1}{NL} \sum_{i=1}^N \sum_{l=1}^L ||f_l^F(\hat{x}_i; \theta), f_l^Q(\hat{x}_i; \theta')||_2. \quad (7)$$

To this end, the total loss for quantization-aware training can be summarized as:

$$\min_{\theta'} \mathcal{L}^Q = \mathcal{L}_{KD} + \beta_{feat} \mathcal{L}_{feat}. \quad (8)$$

# 4    EXPERIMENTS AND RESULTS

In this section, we show the effectiveness of our proposed zero-shot quantization-aware training scheme on MS-COCO 2017 (Lin et al., 2014) and Pascal VOC (Everingham et al., 2010) datasets. Following (Esser et al., 2019), we perform symmetric quantization on both weights and activations across YOLOv5 (Ultralytics, 2021) series and Mask R-CNN (He et al., 2017) for object detection. We also compare our method against standard baselines including LSQ (Esser et al., 2019) and LSQ+ (Bhalgat et al., 2020). Implementation details can be found in Appendix A. We also conduct ablation studies to analyze the effectiveness of different settings and components in Appendix B. In summary, our results establish a state-of-the-art benchmark for zero-shot object detection tasks at different quantization bit-widths while outperforming comparable baselines trained with full real data.

Table 2: Comparison with real data QATs on one-stage YOLOv5 on MS-COCO validation set.

| Method | Real Data | Num Data | Prec. | mAP / mAP50 | | |
|---|---|---|---|---|---|---|
| | | | | YOLOv5-s | YOLOv5-m | YOLOv5-l |
| Pre-trained | ✓ | 120k(full) | FP32 | 37.4/56.8 | 45.4/64.1 | 49.0/67.3 |
| LSQ | ✓ | 120k(full) | | 35.7/54.9 | 43.2/62.2 | 46.0/64.9 |
| LSQ+ | ✓ | 120k(full) | | 35.4/54.6 | 43.3/62.4 | 46.3/64.9 |
| LSQ | ✓ | 2k | W8A8 | 31.6/50.6 | 36.5/55.6 | 40.3/59.1 |
| LSQ+ | ✓ | 2k | | 31.5/50.3 | 36.6/55.8 | 40.1/58.6 |
| **Ours** | × | 2k | | **35.8/55.0** | **43.6/62.3** | **47.3/65.6** |
| LSQ | ✓ | 120k(full) | | 31.5/49.9 | 41.3/60.0 | 43.3/62.1 |
| LSQ+ | ✓ | 120k(full) | | 32.3/50.9 | **41.3/60.3** | 43.4/62.3 |
| LSQ | ✓ | 2k | W6A6 | 28.9/47.2 | 35.0/53.9 | 37.7/55.7 |
| LSQ+ | ✓ | 2k | | 28.6/46.7 | 34.2/52.6 | 37.5/55.8 |
| **Ours** | × | 2k | | **32.7/51.4** | 41.0/59.7 | **45.1/63.3** |
| LSQ | ✓ | 120k(full) | | 26.9/44.9 | 32.9/50.6 | 35.2/53.0 |
| LSQ+ | ✓ | 120k(full) | | 27.0/44.9 | 33.1/51.0 | 35.2/53.4 |
| LSQ | ✓ | 2k | W5A5 | 24.7/42.2 | 31.2/49.3 | 35.2/53.1 |
| LSQ+ | ✓ | 2k | | 25.0/42.9 | 31.2/49.2 | 34.8/52.7 |
| **Ours** | × | 2k | | **28.0/45.8** | **37.1/55.7** | **41.5/59.7** |
| LSQ | ✓ | 120k(full) | | 23.3/40.0 | 27.9/45.4 | 33.1/50.3 |
| LSQ+ | ✓ | 120k(full) | | **23.3/40.2** | 27.7/44.6 | 33.3/50.9 |
| LSQ | ✓ | 2k | W4A4 | 17.2/32.2 | 25.5/42.3 | 28.9/45.7 |
| LSQ+ | ✓ | 2k | | 17.3/32.1 | 26.1/42.6 | 28.6/45.8 |
| **Ours** | × | 2k | | 19.0/33.4 | **29.5/47.1** | **35.0/52.6** |

## 4.1 COMPARISON WITH REAL DATA QATS

To demonstrate the effectiveness of our zero-shot quantization scheme, we select common one-stage object detection networks including YOLOv5-m/s/l as well as two-stage object detection network Mask R-CNN for experimentation and use competitive QAT methods like LSQ (Esser et al., 2019), LSQ+ (Bhalgat et al., 2020) as baselines. Extensive experiments demonstrate that we can outperform both LSQ and LSQ+, which use all real images for training, with only a small amount of ground truth label information. We compare the performance against LSQ and LSQ+ methods, which use 120k/5k real training data of MS-COCO/Pascal VOC dataset. As a comparison, we only use 2k/50 ground truth labels for calibration set generation, the results are presented in Table 2 and Table 3.

**Bit-width** For low-bit-width cases, conventional QAT methods suffer from significant performance degradation, while our method performs good generalization capability. Specifically, for the 4-bit YOLOv5-l case, our method achieves an mAP score of 35.0%, outperforming LSQ by 1.9%. In the 6-bit case, we retain a lead of 1.8% over LSQ. Even in the 8-bit case where LSQ starts to perform well, we still maintain a 1.3% advantage. Similar results were obtained in YOLOv5-s/l.

**Network size** Larger networks tend to exhibit poor performance with existing Quantization-Aware Training (QAT) methods, particularly in low-bit-width cases. For instance, in the 5-bit case, LSQ+ applies to YOLOv5-s resulting in a 10.4% decrease in mAP compared to the pre-trained network, which even achieves 13.8% with YOLOv5-l. In contrast, our approach yields only a 9.4% gap in mAP when quantizing YOLOv5-s to ultra-low 5-bit, and the difference further reduces to 7.5% with YOLOv5-l. Furthermore, as presented in Table 3, we achieve state-of-the-art results with the two-stage object detection network Mask R-CNN, surpassing LSQ trained with full real data at 8-bit width. This further highlights the reliability and versatility of our method.

**Efficiency** QAT methods require the entire training dataset as input, while our method achieves superior results with a condensed synthetic detection calibration set that is only 1/60 of the size of the original. Taking the YOLOv5-m network as an example, LSQ necessitates 120k real images for

Table 3: Comparison with real data QATs on two-stage Mask R-CNN.

| Dataset | Method | Real Data | Num Data | Precision | mAP |
|---|---|---|---|---|---|
| VOC | Pre-trained | ✓ | 5k(full) | FP32 | 75.6 |
| | LSQ | ✓ | 5k(full) | | 72.4 |
| | LSQ | ✓ | 50 | W8A8 | 70.9 |
| | **Ours** | ✗ | 50 | | **72.9** |
| MS-COCO | Pre-trained | ✓ | 120k(full) | FP32 | 38.1 |
| | LSQ | ✓ | 120k(full) | | 35.0 |
| | LSQ | ✓ | 2k | W8A8 | 32.9 |
| | **Ours** | ✗ | 2k | | **35.2** |
| | LSQ | ✓ | 120k(full) | | 34.6 |
| | LSQ | ✓ | 2k | W4A8 | 32.3 |
| | **Ours** | ✗ | 2k | | **34.6** |

Table 4: Comparison with real data PTQs on YOLOv5-s on MS-COCO validation set.

| Network | Method | Real Data | Precision | mAP | mAP50 |
|---|---|---|---|---|---|
| YOLOv5-s | Pre-trained | ✓ | FP32 | 37.4 | 56.8 |
| | LSQ (PTQ only) | ✓ | | 29.5 | 48.2 |
| | LSQ+ (PTQ only) | ✓ | W6A6 | 29.6 | 48.5 |
| | **Ours** | ✗ | | **32.7** | **51.4** |
| | LSQ (PTQ only) | ✓ | | 11.9 | 23.9 |
| | LSQ+(PTQ only) | ✓ | W5A5 | 12.2 | 24.5 |
| | **Ours** | ✗ | | **28.0** | **45.8** |
| | LSQ(PTQ only) | ✓ | | 1.3 | 3.6 |
| | LSQ+(PTQ only) | ✓ | W4A4 | 1.2 | 3.5 |
| | **Ours** | ✗ | | **19.0** | **33.4** |

fine-tuning with 17 minutes per epoch, which takes about 4 hours to converge on two RTX 4090 GPUs. In contrast, our method only requires 2k synthetic calibration samples as input and achieves convergence in approximately 15 minutes, boosting the convergence speed of around 16 times during the fine-tuning phase.

## 4.2 COMPARISON WITH REAL DATA PTQS

We further compare with the post-training quantization (PTQ) using the LSQ/LSQ+ initialization method. We utilize the first batch of data from the MS-COCO 2017 training set as a calibration set to initialize the quantization scaling/bias factors. As presented in Table 4, our method achieves state-of-the-art across different bit-widths, improving the mAP scores by 17.7%/16.2%/3.2% in 4/5/6-bit cases compared to PTQ methods using real data on MS-COCO 2017 validation set. This demonstrates that our synthetic calibration set effectively captures feature information in the advanced MS-COCO dataset.

## 4.3 COMPARISON WITH DATA FREE METHODS

Furthermore, we explore the completely data-free scenario where even sample, label, data distribution are not available and showcase the robustness of our novel adaptive sampling method compared to previous competitors. We compare on quantization-aware training and results are shown in Table 5.

First, we build a weak baseline by adopting **Gaussian** noise as calibration images and find the quantized network fails to converge. This highlights the quality of the calibration set for QAT. We also compare our adaptive sampling method with other proxy datasets. We consider two types —

Table 5: Quantization-aware training results on quantized YOLOv5-s object detector, with a pre-trained full-precision YOLOv5-s as the teacher for knowledge distillation. For a fair comparison, we report results based on 2k synthetic images in all cases. Real Label denotes detailed information about labels including bouding box categories and coordinates. Data Distri. represents quantity distribution information about the labels per image. "-" indicates that the network diverges.

| Network | Prec. | Method | Real label | Data Distri. | mAP | mAP50 |
|---------|-------|--------|------------|--------------|-----|-------|
| YOLOv5-s | FP32 | Baseline(pre-trained) | ✓ | ✓ | 37.4 | 56.8 |
| | W6A6 | Real Label | ✓ | ✓ | 32.7 | 51.4 |
| | | Gaussian noise | × | × | - | - |
| | | Tile(Out-of-distri.) | × | × | 23.9 | 39.0 |
| | | Tile(In-distri.) | × | ✓ | 24.0 | 39.3 |
| | | MultiSample(Out-of-distri.) | × | × | 28.2 | 46.7 |
| | | MultiSample(In-distri.) | × | ✓ | 29.7 | 48.0 |
| | | **Ours(Adaptive Sampling)** | × | × | **32.0** | **50.0** |
| | W5A5 | Real Label | ✓ | ✓ | 28.0 | 45.8 |
| | | Gaussian noise | × | × | - | - |
| | | Tile(Out-of-distri.) | × | × | 16.1 | 27.9 |
| | | Tile(In-distri.) | × | ✓ | 17.7 | 31.0 |
| | | MultiSample(Out-of-distri.) | × | × | 21.9 | 37.3 |
| | | MultiSample(In-distri.) | × | ✓ | 22.5 | 37.4 |
| | | **Ours(Adaptive Sampling)** | × | × | **26.1** | **42.3** |
| | W4A4 | Real Label | ✓ | ✓ | 19.0 | 33.4 |
| | | Gaussian noise | × | × | - | - |
| | | Tile(Out-of-distri.) | × | × | 5.4 | 11.1 |
| | | Tile(In-distri.) | × | ✓ | 6.8 | 13.4 |
| | | MultiSample(Out-of-distri.) | × | × | 11.9 | 22.3 |
| | | MultiSample(In-distri.) | × | ✓ | 13.1 | 23.3 |
| | | **Ours(Adaptive Sampling)** | × | × | **15.0** | **27.0** |

*in-distribution* and *out-of-distribution*. *In-distribution* datasets assume we have quantity distribution information about the labels per image, while *out-of-distribution* datasets suppose we are unaware of the original label. For comparison, we replicate **Tile** method from (Chawla et al., 2021) and implement **MultiSample** method, which directly samples multiple labels for each image randomly. As shown in Table 5, QAT using the images generated by our adaptive sampling outperforms the best *in-distribution* proxy dataset at different bit widths (1.9%/3.6%/2.3% higher at 4-6 bits). Furthermore, We also compared our sampling method with images generated by real labels and observed only a 0.7% difference at 6-bit, which further demonstrates the effectiveness of our data-free sampling method.

## 5 CONCLUSIONS

In this paper, a novel zero-shot quantization framework specially tailored for object detection is proposed. The proposed framework consists of two main components: a novel bounding box and category sampling method for synthetic calibration set generation and a quantization-aware training (QAT) process that incorporates prediction-matching distillation and feature-level distillation to distill knowledge from a pre-trained full-precision network to a quantized network with the synthetic images. Extensive experiments demonstrate that the proposed method is more efficient and accurate than traditional QAT methods like LSQ trained with full real data, empowering ZSQ with immense practical significance for object detection tasks. Moreover, The presented zero-shot adaptive label sampling method for object detection shows significant improvement over other in-distribution proxy datasets and achieves competitive results with real labels. It is obvious that our method can be compatible with any object detection networks with BatchNorm layers including YOLO, Mask R-CNN, *etc.* and the limitation also lies in the inapplicability to networks without BatchNorm layers, which will be explored in the future work.

## 6 REPRODUCIBILITY STATEMENT

The models used in this paper, including various YOLOv5 variants and the Mask-RCNN model, as well as the datasets, such as MS-COCO and VOC, are all open-source. In Appendix A, we provide additional details about the experimental setup, including hardware configurations and hyperparameter settings. In Appendix B, we perform a series of ablation studies on various components of the experimental framework to assess their contributions to the outcomes. Appendix C provides a visual representation of the efficiency advantages of our approach. We believe these supplementary materials will significantly enhance reproducibility.

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

## A IMPLEMENTATION DETAILS

We report mAP and mAP50 on the validation set of MS-COCO 2017(Lin et al., 2014) for the object detection task, utilizing various YOLOv5 variants. All experiments are conducted using pre-trained YOLOv5 as the teacher, and executed on two NVIDIA Geforce RTX 4090 GPUs.

### A.1 ADAPTIVE SAMPLING

While theoretically, merging the updates of labels and images into a single stage seems feasible, our experiments at Section B.1 revealed that the continuously evolving target could detrimentally affect the quality of the generated images. To address this issue, we first conduct a rapid sampling of labels at a low resolution (160) and then use the fixed labels to generate images at a high resolution (640).

### A.2 CALIBRATION SET GENERATION

We apply Eq. 5 and the optimal trade-off parameters for $\{\alpha_{detect}, \alpha_{BN}, \alpha_{TV}, \alpha_{l_2}\}$ are set to $\{0.5, 0.01, 0, 0.0005\}$. We generate $X_{inv}$ by optimizing the cost function for 2500 iterations. We use Adam as the optimizer with an initial learning rate of 1e-2, adjusted by cosine annealing (Loshchilov & Hutter, 2016). We also use cutout (DeVries & Taylor, 2017) as a data augmentation method to enhance the diversity of the synthetic calibration set.

### A.3 QUANTIZATION AWARE TRAINING

Subsequently, we employ the synthesized calibration set for $QAT$. Given the complexity of object detection as a downstream task and the challenges posed by ultra-low-bit quantization, we follow existing literature (Esser et al., 2019), quantizing all layers except the first and last layers. During $QAT$, we use per-tensor symmetric quantization for both activations and weights and learn the quantization scaling/bias factor via back-propagation, with an initial learning rate of 1e-4 in the $ADAM$ optimizer. Rest experimental hyper-parameters follow official YOLOv5[1] implementations. Since $LSQ$ is only evaluated on ImageNet, we re-implement it on YOLO for the object detection task and report mAP/mAP50 as our results. We use Eq. 8 as our loss function, with the optimized hyper-parameters for $\{\beta_{detect}, \beta_{KL}, \beta_{feat}\}$ being $\{0.04, 0.1, 1\}$.

---

[1]https://github.com/ultralytics/yolov5/tree/master

# B ABLATION STUDY

## B.1 ADAPTIVE SAMPLING STAGE

We conduct ablations on the impact of the sampling stage number, results are shown in Table. 6. Overall, the two-stage sampling strategy outperforms the one-stage strategy, which we attribute to the continuous variation of targets causing fluctuation in the regression targets of the image, thus hindering stable convergence. It also matches the performance of the three-stage approach. Ultimately, we opt for the two-stage strategy to strike a balance between performance and cost.

Table 6: Ablations on Adaptive Sampling stages number. One stage: update images and labels simultaneously in one process. Two stages: Relabel first, then synthesize images with fixed labels. Three stages: Generate images with one label first, then relabel with fixed images, and finally synthesize images with fixed labels

| Stages Num | Precision | mAP | mAP50 |
|---|---|---|---|
| One | W6A6 | 30.6 | 48.8 |
| | W5A5 | 25.2 | 41.1 |
| | W4A4 | **16.0** | **27.9** |
| Two | W6A6 | **32.1** | **50.1** |
| | W5A5 | **26.3** | **42.3** |
| | W4A4 | 15.8 | 28.1 |
| Three | W6A6 | 31.7 | 49.3 |
| | W5A5 | 26.1 | 42.5 |
| | W4A4 | 15.7 | 27.8 |

## B.2 CALIBRATION SET SIZE

After hyper-parameters are fixed, the calibration set size $S$ is searched for its optimal trade-off between computation cost and effectiveness with grid search by quantizing YOLOv5-s to 4-8 bits, as displayed in Table 7. When $S$ reaches 2k, the performance of the quantized network approaches convergence. Further increasing the size will lead to increased data generation time and computational costs. To avoid complex searches, $S$ is used for all experiments. While this may not be optimal for all networks, it is sufficient to demonstrate the superiority of our approach.

Table 7: A detailed analysis of calibration set size $S$ across different bit widths

| Method | Real Data | $S$ | mAP | | | | |
|---|---|---|---|---|---|---|---|
| | | | W4A4 | W5A5 | W6A6 | W7A7 | W8A8 |
| LSQ | ✓ | 120k (Full) | 23.3 | 26.9 | 31.5 | 33.4 | 35.7 |
| Ours | × | 5k | 19.1 | **28.0** | 32.6 | 34.9 | 35.7 |
| | × | 4k | 18.9 | 27.9 | **32.8** | 34.7 | 35.8 |
| | × | 3k | **19.2** | 27.9 | 32.7 | **35.0** | **36.0** |
| | × | 2k | 19.0 | 27.4 | 32.7 | 34.7 | 35.4 |
| | × | 1k | 18.3 | 27.8 | 32.6 | 34.8 | 35.6 |

## B.3 MODULES

Ablation on key modules of the QAT stage including $\mathcal{L}_{KL}$ (Kullback-Leibler Loss, Eq. 6), $\mathcal{L}_{detect}$ (Eq. 6), and $\mathcal{L}_{feat}$ (Eq. 7) is conducted. As presented in Table 8, dropping one or two of them results in a mAP loss. The largest mAP loss (7.2%) occurs when both $\mathcal{L}_{KL}$ and $\mathcal{L}_{feat}$ are removed, indicating their cooperative relationship: $\mathcal{L}_{feat}$ constrains features of network layers, facilitating $\mathcal{L}_{KL}$ to align the network's predictions with the targets.

Table 8: Ablations on modules. We use 2k calibration set and report mAP/mAP50 of 4-bit YOLOv5-s on MS-COCO validation set.

| $\mathcal{L}_{feat}$ | $\mathcal{L}_{KL}$ | $\mathcal{L}_{detect}$ | mAP | mAP50 |
|---|---|---|---|---|
| ✓ | ✓ | ✓ | **19.0** | **33.4** |
| | ✓ | ✓ | 16.8 | 30.1 |
| | | ✓ | 11.8 | 21.5 |

(a) Number of images involved in different training sets.

(b) Quantization-aware training (QAT) convergence time on 5-bit YOLOv5-m.

Figure 4: (a) Our synthetic condensed calibration set is $60\times$ smaller than the MS-COCO training set. (b) The training convergence speed can be improved by up to $16\times$ compared to LSQ.

## C  SAMPLE EFFICIENCY

We would like to emphasize that by employing **Adaptive Sampling**, we achieved comparable or even superior results on QAT using a synthetic calibration set that is only **1/60** the size of the original training dataset. Additionally, by integrating self-distillation into the fine-tuning process of the quantized object detection network, we enabled a more efficient knowledge transfer. In the initial stage, utilizing 8 RTX 4090 GPUs for image generation allows us to produce 256 images every 20 minutes, resulting in a total of 160 minutes to generate 2,000 images. It is important to note that the calibration set we generate captures the overall characteristics of the original training set, allowing it to be reused multiple times during the quantization-aware training process. As the number of training iterations increases, our method progressively enhances the training convergence speed, achieving up to **16x** faster convergence compared to the LSQ method trained on the full real dataset. The corresponding results are visually illustrated in Fig. 4.

## D  ADDITIONAL QUALITATIVE RESULTS

**Qualitative results for synthetic data**   In this section, we highlight the advantages of our Adaptive Sampling method over both random sampling for multiple labels and the False Positive Sampling approach proposed by (Chawla et al., 2021). As shown in Fig. 5, the left side illustrates our Adaptive Sampling method, which initially starts with single-label random sampling as presented in Table 1. After Adaptive Sampling, the model leverages the information stored during pre-training and add objects it considers highly confident, ultimately producing high-quality images. For instance, you can

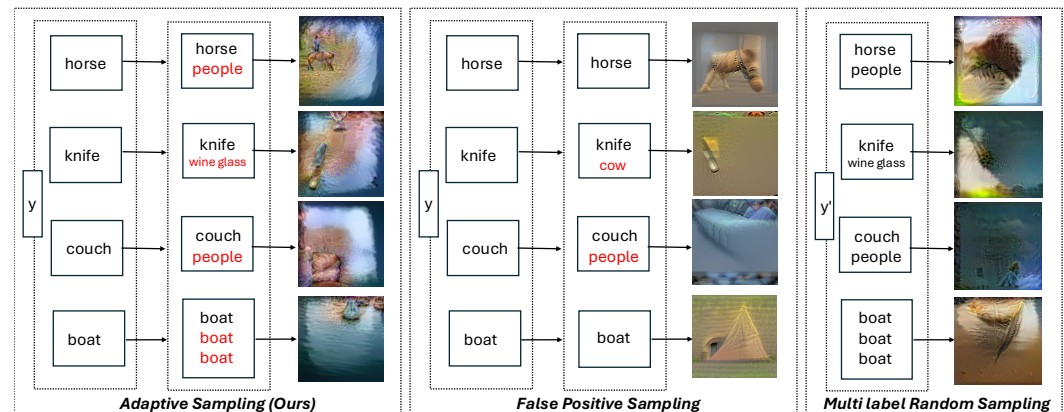

Figure 5: A comparison of the image quality generated by various sampling methods.

observe a person riding a horse, three boats gently floating on the shimmering water, and someone about to sit and rest next to a couch, etc.

Next, we use the obtained labels to perform multi-label random sampling by generating the corresponding object sizes and locations based on the sampling distribution in Table 1. The resulting images are shown on the right side of Fig. 5. In this scenario, the image quality deteriorates significantly, and the visual features fail to accurately reflect the generated objects. Consequently, compared to multi-label sampling, our Adaptive Sampling method captures the model's internal information more effectively, producing higher-quality and more coherent images.

Additionally, in the middle part of Fig. 5, we use the same single-label input to generate data using the false-positive sampling method from (Chawla et al., 2021). Compared to our adaptive sampling method, the false-positive sampling approach often fails to synthesize additional objects beyond the initial target in certain cases. Moreover, the overall quality of the generated images is noticeably lower than that achieved with our method. This further demonstrates the effectiveness of our sampling approach.

**Qualitative results for object detection performance**    In this section, we present visualizations showcasing the object detection capabilities of various neural networks. Specifically, we randomly selected four images from the COCO validation set and used the detection results of a full-precision YOLOV5-s network as the reference. The visual comparisons display the detection results of neural networks trained using our adaptive sampling method versus those trained with the false positive sampling method from (Chawla et al., 2021) on 4-bit quantization-aware training (QAT). The results are shown in Fig. 6.

The visualizations reveal that our quantized network can detect objects that the network trained with false positive sampling fails to recognize, such as teddy bears and toilets. Furthermore, in scenarios where only one object is present, our quantized network demonstrates higher confidence (0.72), outperforming the other quantized network by 9% in confidence level.

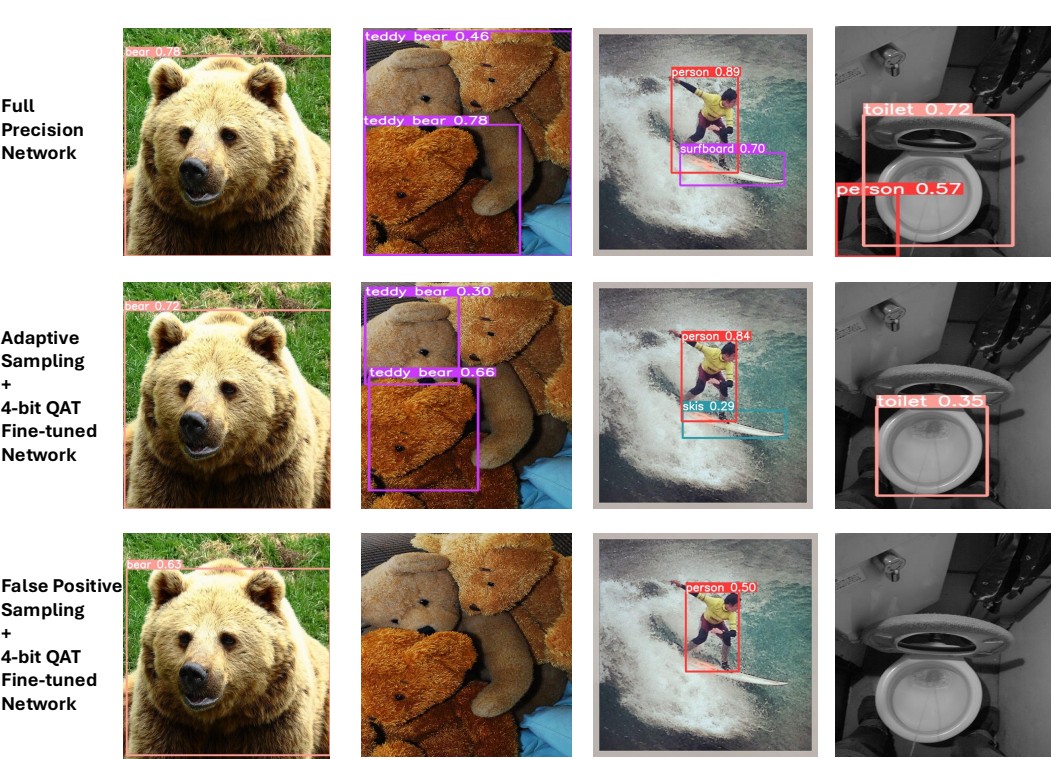

Figure 6: Qualitative analysis of object detection performance across different neural networks

