# OpenReview forum: "Zero-shot Quantization for Object Detection"
_ICLR.cc/2025/Conference — Submitted to ICLR 2025_

### Official Review · Reviewer_R2o8 · 2024-10-31

**Soundness:** 2
**Presentation:** 2
**Contribution:** 3
**Rating:** 5
**Confidence:** 3

**Summary:**

This work only uses the pre-trained object detection network to reconstruct the training samples without accessing any original data or labels, and realizes zero-shot quantization of the object detection network.

**Strengths:**

1. The problem raised by the paper is clearly expressed and the motivation is clearly stated.
2. This paper demonstrates the effectiveness of the proposed method through the extensive experiments.

**Weaknesses:**

Although the paper has demonstrated the effectiveness of the method in experiments, the explanation of the proposed method is difficult to follow.
1. In Section 3.2 and ‘Relabel’ paragraph, The supervision mentioned in formula 5 is done by GT. In my understanding, the role of GT is replaced by the generation of the "teacher" model. In the initial stage, all information such as image and position are initialized as Gaussian noise. How does the teacher model obtain information to complete this optimization? I still feel unclear about the establishment process of this optimization.
2. After this sentence, the author only mentioned that this is the 'teacher' model. Is it the model itself that is being quantized or is it a model with other requirements? The author did not elaborate.
3. It is recommended that the author add a detailed explanation of f_l after formula 7. What specific layers does the author distill?
4. In appendix C, the author only compared the amount of data and the cost of the training process. What is the time consumption of generating samples in the first stage?

**Questions:**

In page 1, the paper mentions that ‘Classification networks require only a randomly sampled category id label for data synthesizing’, which means that classification task also samples the gt labels. Why can not detection task methods take gt as label for location and size generation?

---

> ### Author Response · Authors · 2024-11-22
>
> We would like to thank Reviewer R2o8 for the insightful feedback. Your suggestions for clarification and improvement are constructive. Please find the responses below:
>
> **Q1: In Section 3.2 and ‘Relabel’ paragraph, The supervision mentioned in formula 5 is done by GT. In my understanding, the role of GT is replaced by the generation of the "teacher" model. In the initial stage, all information such as image and position are initialized as Gaussian noise. How does the teacher model obtain information to complete this optimization?**
>
> A1: Thank you for your question. **The teacher model is fixed throughout the process and does not obtain any information. Instead, we optimize the initialized Gaussian noise image and GT to obtain the object information from the teacher model, including the object category, the box coordinates, and the category distribution.** In Formula 5, x represents the trainable input image, and y is the ground truth label information, which includes label type, size, and location details. ϕ(x) denotes the neural network’s predicted label information for the image.
>
> In the "Relabel" paragraph of Section 3.2, we start by independently sampling a label set 𝑌 for each image based on the sampling method outlined in Table 1, initializing the input image 𝑋 as Gaussian noise. In this context, 𝑋 in Formula 5 represents Gaussian noise, while 𝑌 denotes the initially sampled single label set. We then calculate the gradient of 𝑋 and update it to minimize the loss defined in Formula 5, keeping the label 𝑌 fixed. The terms $L_{BNS}$ and $L_{prior}$ guide 𝑋 toward characteristics of real data used in model training, while $L_{detect}$ drives ϕ(x) to predict labels closer to 𝑌.
>
> After several rounds of updates, 𝑋 starts to acquire basic image features and capture information from 𝑌. At this point, we use a pretrained network to recognize 𝑋 and incorporate high-confidence regions into 𝑌 while removing low-confidence  labels, thereby forming an updated Y. This iterative process continuously refines both the image 𝑋 and the label set 𝑌 until convergence. The resulting synthetic images effectively capture real image characteristics from the pre-trained teacher model, providing high-quality synthetic data for QAT training.
>
> **Q2: Is it the model itself that is being quantized or is it a model with other requirements?**
>
> A2: Thank you for the question. **The teacher and student models represent the models before and after quantization, respectively.** Thus, the teacher model refers to the quantized model in its full-precision form. A visual representation can be found in Figure 2, where the "Full-precision network" corresponds to the full-precision teacher model, and the "Quantized network" corresponds to the quantized student model. Both models share the exact same network architecture, with the only difference being that the teacher model’s parameters have higher numerical precision.
>
> **Q3: It is recommended that the author add a detailed explanation of f_l after formula 7. What specific layers does the author distill?**
>
> A3: Thank you for the suggestion. Our key models consist of the CNN-based single-stage object detection model YOLO and the CNN-based two-stage model Mask-RCNN. In these models, the CNN layers are primarily responsible for feature extraction and combination, with parameters that capture substantial information from the features of training images. Similarly, the Batch Normalization (BN) layers retain scaling (γ) and shifting (β) coefficients derived from training data. Therefore, in our experiments, we selected fl to include all CNN and BN layers, allowing for a more comprehensive extraction of information about the real training data from the pre-trained model. Additionally, we observed that choosing fl as only the CNN layers still yielded comparable results, likely because the CNN layers contain significantly more parameters than the BN layers. This suggests that sufficient information can be distilled from the CNN layers alone to enhance the model's performance on real-world images.
>
> **Q4: What is the time consumption of generating samples in the first stage?**
>
> A4: Thanks for the question. In the initial stage, utilizing 8 RTX 4090 GPUs for image generation allows us to produce 256 images every 20 minutes, resulting in a total of 160 minutes to generate 2,000 images. Additionally, the calibration set we generate effectively captures the overall characteristics of the training set and can be reused multiple times throughout the model's quantization-aware training process. As the number of training iterations increases, our method's acceleration rate can ultimately achieve a 16x speedup during the training phase.  **We also include the time consumption in Appendix C of the revised version of manuscript.**

---

> > ### Author Response · Authors · 2024-11-22
> >
> > **Q5: Why can not detection task methods take gt as label for location and size generation?**
> >
> > A5: Please refer to line 260, where we present the several challenges in detection gt sampling compared with classification tasks. First, the location and size of objects within the network are unknown. The labels for object detection networks include both object categories and bounding boxes, and randomly sampling box and categories faces challenges related to the plausibility of the relative positions and sizes of the objects. For example, it is unreasonable for an elephant to be on a plane. As shown in Section B.4, the performance of random sampling using given multi-object labels is significantly worse than that of our adaptive sampling method. Besides, object categories distribution is also unknown and thus we can't manually arrange labels according to plausibility (We have no way of knowing which category is an airplane or an elephant in zero-shot constraint). Current method typically uses ground truth (both real categories and bounding boxes) to guide model inversion, which is not truly zero-shot.

---

> > > ### Author Response · Authors · 2024-11-25
> > > **Gentle reminder: The interactive discussion period will end in less than two days**
> > >
> > > Dear Reviewer R2o8,
> > >
> > > Thank you again for your time and efforts in reviewing our paper.
> > >
> > > As the discussion period draws close, we kindly remind you that two days remain for further comments or questions. We would appreciate the opportunity to address any additional concerns you may have before the discussion phase ends.
> > >
> > > Thank you very much.
> > >
> > > Best regards,
> > >
> > > Authors

---

> > > > ### Comment · Reviewer_R2o8 · 2024-11-26
> > > >
> > > > Thank you for your time and responses.
> > > > For A1, I am positive about the novelty of the author's motivation, but I am still confused by the author's description of the method in Figure 2. I can't find L_{BNS} and L_{TV} in eq.4 and eq.5 in this overview figure. And this figure show L_{KL} but I only find L_{KD} in eq.6 and eq.8.
> > > > For A4, if the method needs 8 RTX 4090 GPUs and 160 minutes to generate for YOLOv5-s, then for some of the larger detection networks that need to be quantized, is this generation pressure too great?

---

> > > > > ### Author Response · Authors · 2024-11-26
> > > > >
> > > > > **Q1:  For A1, I am positive about the novelty of the author's motivation, but I am still confused by the author's description of the method in Figure 2. I can't find $L_{BNS}$ and $L_{TV}$ in eq.4 and eq.5 in this overview figure. And this figure show $L_{KL}$ but I only find $L_{KD}$ in eq.6 and eq.8.**
> > > > >
> > > > > A1: We sincerely appreciate the reviewer’s thorough feedback and are grateful for your recognition of the novelty of our motivation.
> > > > >
> > > > > Regarding $L_{BNS}$ and $L_{TV}$, these loss functions are commonly used in zero-shot data generation. Specifically, $L_{BNS}$ is designed to optimize the inputs to align with the batch normalization (BN) statistics of the target recognition network. For instance, full-precision recognition networks like YOLO typically store the running mean and variance parameters of real training data in their BN layers during pretraining. $L_{BNS}$ minimizes the distance between the statistics (mean and variance) of the inputs and these stored parameters across all BN layers, thereby enforcing feature similarity between the synthetic and real images.
> > > > >
> > > > > On the other hand, $L_{TV}$ ensures the smoothness of the synthetic images by minimizing the Frobenius norm of differences between adjacent pixels. To enhance clarity for readers, we have annotated the relevant details for $L_{BNS}$ and $L_{TV}$ in Stage 1 of Figure 2. In this context, $L_{BNS}$ involves all the BN layers in the network, while $L_{TV}$ is computed solely based on the pixel values of the inputs.
> > > > >
> > > > > Additionally, we apologize for the mistake in Stage 2 of Figure 2, where $L_{KD}$ was incorrectly labeled as $L_{KL}$. We thank the reviewer for pointing this out and have corrected it in the revised version.
> > > > >
> > > > > **Q2: For A4, if the method needs 8 RTX 4090 GPUs and 160 minutes to generate for YOLOv5-s, then for some of the larger detection networks that need to be quantized, is this generation pressure too great?**
> > > > >
> > > > > A2:  Thank you for your insightful question. Data generation tasks typically involve significant resource overhead, which is often unavoidable. However, our work has reduced the sample size for data generation to a manageable level (2k) while achieving results comparable to using the full real dataset (120k).
> > > > >
> > > > > Our goal is to provide insights from a novel zero-shot perspective, and we recognize that accelerating the data synthesis process is a promising direction for future research. Notably, we have observed that a recent paper in the ViT domain has already begun exploring methods to speed up this process. [1] Once again, we appreciate the reviewer’s valuable and constructive feedback.
> > > > >
> > > > > Reference:
> > > > >
> > > > > [1] Hu, Zixuan, Yongxian Wei, Li Shen, Zhenyi Wang, Lei Li, Chun Yuan, and Dacheng Tao. "Sparse Model Inversion: Efficient Inversion of Vision Transformers for Data-Free Applications." In *Forty-first International Conference on Machine Learning*.

---

### Official Review · Reviewer_f6ut · 2024-11-02

**Soundness:** 3
**Presentation:** 3
**Contribution:** 2
**Rating:** 5
**Confidence:** 4

**Summary:**

This paper introduces a novel zero-shot quantization (ZSQ) framework designed specifically for object detection tasks. Unlike existing ZSQ approaches that are mainly suited for classification, this framework tackles the unique challenges of object detection, including localization and class imbalance. The framework involves two main components: (1) a bounding box and category sampling method for generating a synthetic calibration set, which approximates the spatial and categorical distribution of the objects without any real data, and (2) a quantization-aware training (QAT) process that integrates feature-level distillation for effective knowledge transfer. The proposed method achieves impressive results, outperforming baselines like LSQ on MS-COCO and Pascal VOC with only 1/60 of the data size typically required.

**Strengths:**

1. The bounding box and category sampling strategy provide a data-efficient way to approximate real-world object distributions, making it suitable for privacy-sensitive applications.
2. Integrating feature-level distillation into the QAT stage improves knowledge transfer and performance in low-bit quantization settings, significantly narrowing the performance gap between full-precision and quantized networks.
3. The framework demonstrates superior performance on both MS-COCO and Pascal VOC datasets, even in ultra-low-bit scenarios, thus highlighting its robustness and practical utility.
4. By requiring only a fraction of the typical calibration data size, the method enhances scalability for deployment on resource-constrained devices.

**Weaknesses:**

1. The framework combines synthetic data generation, adaptive sampling, and feature-level distillation within quantization-aware training (QAT), creating a complex pipeline. This complexity may hinder practical adoption and implementation in real-world settings.
2. The framework relies heavily on the quality of synthetic data for calibration, which may introduce variability in performance, especially in challenging real-world environments. Ensuring consistency in synthetic data quality is critical for stable results.
3. While the paper demonstrates strong results with YOLOv5 and Mask R-CNN, it lacks testing across a wider range of object detection architectures. Broader experimentation would strengthen confidence in the framework's generalizability.
4. The method's reliance on synthetic category distribution for handling class imbalance may not fully capture the complexities of real-world distributions. This could impact performance in scenarios with highly skewed class distributions.
5. Although the paper suggests that a small calibration set suffices, a more granular analysis of the calibration set size across different bit-widths could provide insights into optimal data requirements and help streamline data efficiency.
6. The method's effectiveness largely depends on feature-level and prediction-matching distillation. Over-reliance on distillation may limit its flexibility for future developments in quantization techniques that don’t use distillation.

**Questions:**

Please refer to the Weaknesses box.

---

> ### Author Response · Authors · 2024-11-22
>
> We sincerely appreciate Reviewer f6ut for the insightful feedback. Your suggestions and insights are very helpful for further enhancing the submission quality. Please find the responses below:
>
> **Q1: The framework combines synthetic data generation, adaptive sampling, and feature-level distillation within quantization-aware training (QAT), creating a complex pipeline. This complexity may hinder practical adoption and implementation in real-world settings.**
>
> A1: Thank you for the suggestion. Our experimental setting focuses on quantization-aware training (QAT) of object detection models in a zero-shot scenario. In object detection tasks, relevant information includes label, size, and location distributions. As demonstrated in the ablation study in Table 5, the absence of any distribution information (Gaussian distribution) or the use of only label distribution (as in in-distribution MultiSample and Tile) results in a significant decline in QAT performance. In contrast, our proposed adaptive sampling method effectively extracts all three types of distribution information from the pretrained model. This emphasizes the critical role of our adaptive sampling within the entire pipeline.
>
> Additionally, data generation and knowledge distillation are also standard paradigms in zero-shot tasks, as seen in [1], [2], [3], [4]. We incorporate more fine-grained, feature-level distillation in the knowledge distillation phase, as QAT requires more precise control than full-precision fine-tuning to achieve optimal performance.
>
> **Q2: The framework relies heavily on the quality of synthetic data for calibration, which may introduce variability in performance, especially in challenging real-world environments. Ensuring consistency in synthetic data quality is critical for stable results.**
>
> A2: Thank you for the suggestion. To further demonstrate the robustness and superior performance of our method, we conducted additional experiments. Starting with Gaussian noise, we utilized our adaptive sampling approach to generate 5,000 synthetic images and randomly selected 2,000 images for each quantization-aware training session. Using YOLO v5-s as the base model, we evaluated performance under various bit-width configurations, repeating each experiment five times. The performance of each run, along with the standard deviation across results, is presented in Table A.
>
> As illustrated in Table A, despite variations in the synthetic datasets used for each run, the results consistently demonstrate stable and reliable performance, underscoring the robustness of our method when applied to diverse synthetic data inputs.
>
> **Table A Method Stability: Comparison Using Different Synthetic Data for QAT Training (YOLOv5-s on MS-COCO)**
>
> |                    | W4A4  | W5A5  | W6A6  |
> | ------------------ | ----- | ----- | ----- |
> | SEED=0             | 19.1  | 27.9  | 32.5  |
> | SEED=21            | 18.7  | 27.9  | 32.7  |
> | SEED=42            | 18.8  | 28.2  | 32.6  |
> | SEED=63            | 18.5  | 28.0  | 32.6  |
> | SEED=84            | 19.3  | 27.7  | 32.6  |
> | Standard Deviation | 0.286 | 0.162 | 0.063 |
>
> > We collected a dataset of 5,000 synthetic samples and randomly sampled 2,000 samples each time to perform QAT training. The results, including standard deviations, are reported under different bit-width configurations. Here, **WXAX** indicates that both weights and activations are quantized to X bits during the QAT training process.
>
>
>
> Reference:
>
> [1] Zhong, Yunshan, et al. "Intraq: Learning synthetic images with intra-class heterogeneity for zero-shot network quantization." Proceedings of the IEEE/CVF Conference on Computer Vision and Pattern Recognition. 2022.
>
> [2] Liu, Yuang, Wei Zhang, and Jun Wang. "Zero-shot adversarial quantization." Proceedings of the IEEE/CVF Conference on Computer Vision and Pattern Recognition. 2021.
>
> [3] Li, Huantong, et al. "Hard sample matters a lot in zero-shot quantization." Proceedings of the IEEE/CVF conference on Computer Vision and Pattern Recognition. 2023.
>
> [4] Zhong, Yunshan, et al. "Fine-grained data distribution alignment for post-training quantization." European Conference on Computer Vision. Cham: Springer Nature Switzerland, 2022.

---

> > ### Author Response · Authors · 2024-11-22
> >
> > **Q3: While the paper demonstrates strong results with YOLOv5 and Mask R-CNN, it lacks testing across a wider range of object detection architectures. Broader experimentation would strengthen confidence in the framework's generalizability.**
> >
> > A3: Thank you for your insightful comments and for highlighting the potential need for broader experimentation across different object detection architectures in our paper. We have carefully considered the selection of YOLOv5 and Mask R-CNN for our experiments, as these models represent a diverse range of detection paradigms. YOLOv5 exemplifies single-shot detectors known for their speed and efficiency, while Mask R-CNN represents a more complex, two-stage approach. These two models have been widely adopted and serve as benchmarks within the computer vision community, making them representative of the state-of-the-art in object detection. We believe that the strong results demonstrated with these models provide a solid foundation for the effectiveness of our framework. Our current study is constrained by time limitations which prevent us from incorporating other architectures into our analysis at this stage though we understand the importance of generalizability and will address this in the discussion section of our paper by outlining potential future work to extend our framework to other architectures.
> >
> > **Q4: The method's reliance on synthetic category distribution for handling class imbalance may not fully capture the complexities of real-world distributions. This could impact performance in scenarios with highly skewed class distributions.**
> >
> > A4: We appreciate the question and would like to clarify that the COCO dataset used in this study is a real-world dataset characterized by a highly imbalanced class distribution. For instance, frequently occurring labels such as “person” and “car” appear 10,777 and 1,918 times, respectively, in the COCO validation set, whereas less common labels like “toaster” and “teddy bear” appear only 9 and 11 times, respectively. This creates a frequency disparity among labels that can exceed a factor of 1,000.
> >
> > Despite this significant imbalance, our method achieves strong results on the validation set, demonstrating its robustness and ability to maintain stable performance even when applied to datasets with highly skewed class distributions.
> >
> > **Q5: Although the paper suggests that a small calibration set suffices, a more granular analysis of the calibration set size across different bit-widths could provide insights into optimal data requirements and help streamline data efficiency.**
> >
> > A5: We appreciate the Reviewer's insightful suggestion. To address this, we have conducted an ablation study on the impact of calibration set size across different bit-widths. Specifically, we utilized the YOLOv5-s model and performed quantization-aware training with bit-widths ranging from 4 to 8, using calibration set sizes varying from 1k to 5k samples. The results are presented in Table B.
> >
> > **Table B A More Detailed Analysis of Calibration Set Sizes Across Different Bit Widths (YOLOv5-s on MS-COCO)**
> >
> > | Precision →            | W4A4     | W5A5     | W6A6     | W7A7     | W8A8     |
> > | :--------------------- | :------- | -------- | -------- | -------- | -------- |
> > | Calibration Set Size ↓ | mAP ↓    | mAP ↓    | mAP ↓    | mAP ↓    | mAP ↓    |
> > | 5k                     | 19.1     | **28.0** | 32.6     | 34.9     | 35.7     |
> > | 4k                     | 18.9     | 27.9     | **32.8** | 34.7     | 35.8     |
> > | 3k                     | **19.2** | 27.9     | 32.7     | **35.0** | **36.0** |
> > | 2k                     | 19.0     | 27.4     | 32.7     | 34.7     | 35.4     |
> > | 1k                     | 18.3     | 27.8     | 32.6     | 34.8     | 35.6     |
> >
> > As shown, when the calibration set size reaches 2k, the performance of the quantized network stabilizes at a relatively converged level. While 2k may not be the optimal choice for all network architectures, it is sufficient to effectively demonstrate the superiority of our approach. **We also include the detailed analysis in Appendix B.2 of the revised version of manuscript.**

---

> > > ### Author Response · Authors · 2024-11-22
> > >
> > > **Q6: The method's effectiveness largely depends on feature-level and prediction-matching distillation. Over-reliance on distillation may limit its flexibility for future developments in quantization techniques that don’t use distillation.**
> > >
> > > A6: Thank you for the suggestion. First, our method’s effectiveness depends not only on knowledge distillation but also significantly on the quality of sampled images. For example, in Table 5, we use Gaussian noise as input in an ablation study to assess the impact of image quality. In this experiment, we keep the framework’s fine-tuning and knowledge distillation setup unchanged, with image quality as the only variable. The resulting QAT-trained network fails to converge, indicating that our adaptive sampling method effectively extracts data characteristics from the pretrained model and confirming that the effectiveness of our method is not solely due to distillation.
> > >
> > > Additionally, for the Tile and MultiSample baselines in Table 5, while their label distributions align with the original COCO dataset in the in-distribution setting, their object size and location distributions differ from the original data, leading to a drop in accuracy. This finding suggests that label, size, and location distributions are all critical for achieving accurate QAT training.
> > >
> > > Furthermore, in zero-shot quantization (ZSQ) scenarios, using only the object detection loss without knowledge distillation can lead to catastrophic results, as shown in Table 8. Therefore, the use of knowledge distillation in ZSQ has become a standard approach; related works, such as [1][2][3][4], all leverage knowledge distillation to support model training.
> > >
> > > Reference:
> > >
> > > [1] Zhong, Yunshan, et al. "Intraq: Learning synthetic images with intra-class heterogeneity for zero-shot network quantization." Proceedings of the IEEE/CVF Conference on Computer Vision and Pattern Recognition. 2022.
> > >
> > > [2] Liu, Yuang, Wei Zhang, and Jun Wang. "Zero-shot adversarial quantization." Proceedings of the IEEE/CVF Conference on Computer Vision and Pattern Recognition. 2021.
> > >
> > > [3] Li, Huantong, et al. "Hard sample matters a lot in zero-shot quantization." Proceedings of the IEEE/CVF conference on Computer Vision and Pattern Recognition. 2023.
> > >
> > > [4] Zhong, Yunshan, et al. "Fine-grained data distribution alignment for post-training quantization." European Conference on Computer Vision. Cham: Springer Nature Switzerland, 2022.

---

> > > > ### Author Response · Authors · 2024-11-25
> > > > **Gentle reminder: The interactive discussion period will end in less than two days**
> > > >
> > > > Dear Reviewer f6ut,
> > > >
> > > > Thank you again for your time and efforts in reviewing our paper.
> > > >
> > > > As the discussion period draws close, we kindly remind you that two days remain for further comments or questions. We would appreciate the opportunity to address any additional concerns you may have before the discussion phase ends.
> > > >
> > > > Thank you very much.
> > > >
> > > > Best regards,
> > > >
> > > > Authors

---

### Official Review · Reviewer_3tqE · 2024-11-02

**Soundness:** 2
**Presentation:** 2
**Contribution:** 2
**Rating:** 5
**Confidence:** 3

**Summary:**

This paper proposes a method for solving the problem of Network Quantization for Object Detection in the zero-shot setting. The contribution of this paper is an adaptive sampling method (Relabel), which is used in the Synthetic data generation phase of the zero-shot setting. The main idea of Relabel is to use the prediction from the pretrained full-precision detection model to refine the bounding box/ class label during the optimization of Synthetic data generation. With the adaptive sampling strategy, the model performance can be improved while using less synthetic data than the baselines.

**Strengths:**

- Except for the points pointed out in the weakness section, this paper has a clear structure, and running the experiments/ablation study to validate the effectiveness of the proposed methods.

- The idea of Relabel is good, and can effectively improve the generation quality of the Zero-shot Quantization method.

- The paper claims this strategy can reduce the number of samples, thus reducing the training time, which is essential for the Quantization problem.

**Weaknesses:**

**Clarity**: There are some claims that need to clarify for the experiment/qualitative/quantitative analysis:
-  In L78 and L79, the paper claims category imbalance in the current detection dataset, can you provide some examples illustrating that the previous method is struggling with it? Also, can you provide visualization/analysis your method is capable of handling the imbalance problems? (For example, using some metric to show the class distribution of the generated samples of the previous method and your method.)

- In L80 and L81, the paper claims "fine-tuning strategy for zero-shot quantized detection network with synthetic calibration data has not been studied". Can the author clarify more about this claim, since [1] also fine-tuning from synthetic data? What is the difference between your fine-tuning approach compared to [1] and how you can improve from [1]?

**Experiment**:
- In the main paper, the paper should include the qualitative results of the synthetic data, compared with the baseline and [1], and also include the qualitative results from the quantized network and the full-precision network.
- Table 5, why the False positive sample from [1] is not included?
- What is the purpose of the weak baseline Gaussian in Table 5?


[1] Akshay Chawla, Hongxu Yin, Pavlo Molchanov, and Jose Alvarez. Data-free knowledge distillation for object detection. In Proceedings of the IEEE/CVF Winter Conference on Applications of Computer Vision, pp. 3289–3298, 2021.

**Questions:**

All concerns and questions are listed in the Weakness section.

---

> ### Author Response · Authors · 2024-11-22
>
> Thank you for your detailed suggestions and comments. Please find the corresponding responses below:
>
> **Q1: In L78 and L79, the paper claims category imbalance in the current detection dataset, can you provide some examples illustrating that the previous method is struggling with it? Also, can you provide visualization/analysis your method is capable of handling the imbalance problems?**
>
> A1: Thank you for the suggestion. First, we want to clarify that the current zero-shot quantization studies can only generate balanced datasets, as seen in studies like [2],[3], [4], [5]. However, in real-world scenarios, class distributions in object detection tasks are often highly imbalanced. For example, as illustrated in Figure 1(b), labels such as “person” and “car” are very common, whereas labels like “hair dryer” and “toaster” appear much less frequently. Our method can generate a label distribution that closely matches the original label distribution, as visualized in Figure 1(b). To quantify the difference between label distributions sampled by different methods and the original label distribution, we use KL divergence as a metric (normalizing all distributions so that the sum of label probabilities equals 1).
>
> **Table A: Comparison of KL Divergence Across Various Label Distributions**
>
> $KL(P_{ours}||P_{data}) $ : **0.177**
>
> $ KL(P_{fp}||P_{data}) $ : 0.241
>
> $KL(P_{ms}||P_{data})$ : 0.614
>
> $KL(P_{tile}||P_{data})$ : 0.630
>
> Here, $P_{data}$ refers to the label distribution of the original COCO training set, while$P_{ours}$, $P_{fp}$, $P_{ms}$, and $P_{tile}$ represent the label distributions obtained by sampling 2048 data points using our adaptive sampling method, as well as the false positive sampling, random multi-label sampling, and tile sampling methods from [1], respectively. As shown, our adaptive sampling method achieves the smallest KL divergence compared to all other methods, indicating that it most effectively preserves the characteristics of the original data.
>
>
>
> **Q2: In L80 and L81, the paper claims "fine-tuning strategy for zero-shot quantized detection network with synthetic calibration data has not been studied". Can the author clarify more about this claim, since [1] also fine-tuning from synthetic data? What is the difference between your fine-tuning approach compared to [1] and how you can improve from [1]?**
>
> A2: We appreciate your insightful question regarding the differences in the fine-tuning phase between our work and [1].
>
> This claim highlights that no prior work in the field of object detection has explored network quantization in a data-free setting. Although [1] also uses synthetic data for fine-tuning, there are two key differences in its approach compared to ours.
>
> First, in the framework of [1], the student network remains a full-precision network, with weights updated using full-precision gradient information. In contrast, we train a quantized model through knowledge distillation using Straight-Through Estimator (STE), which is a more challenging approach. Second, we employ more fine-grained distillation information: while [1] relies solely on final logits distillation, we combine feature maps from intermediate layers with the final layer logits for distillation.
>
> In our quantization-aware training setting, using only the fine-tuning method from [1] leads to a notable performance drop, as shown in Table B.
>
> **Table B Comparison with Logits Distillation using the same synthetic data (YOLOv5-s on MS-COCO)**
>
> | Method                  | Real Data | Num Data    | Precision | mAP      |
> | :---------------------- | :-------- | :---------- | :-------- | :------- |
> | Pre-trained             | √         | 120k (full) | FP32      | 37.4     |
> | Logits Distillation [1] | ×         | 2k          | W6A6      | 31.8     |
> | **Our Distillation**    | ×         | 2k          |           | **32.6** |
> | Logits Distillation     | ×         | 2k          | W5A5      | 26.9     |
> | **Our Distillation**    | ×         | 2k          |           | **28.1** |
> | Logits Distillation     | ×         | 2k          | W4A4      | 16.8     |
> | **Our Distillation**    | ×         | 2k          |           | **18.6** |

---

> > ### Author Response · Authors · 2024-11-22
> >
> > **Q3: In the main paper, the paper should include the qualitative results of the synthetic data, compared with the baseline and [1], and also include the qualitative results from the quantized network and the full-precision network.**
> >
> > A3: Thank you for the suggestion. In the updated manuscript (Appendix D), we have added additional qualitative results. Specifically, we compare data synthesized using our adaptive sampling method with data generated by the false-positive sampling method from [1] and random multi-label sampling as baseline. These comparisons visually demonstrate that our sampling method produces higher-quality data and captures more label information from the object detection network.
> >
> > Furthermore, we compare the performance of networks trained with QAT using data synthesized by adaptive sampling versus those trained with data generated by false-positive sampling on real images. By comparing the results with those of a full-precision network, it is evident that our synthesized data more effectively preserves the object detection capabilities of the network under low-precision settings. **This result shows that the distillation task and the quantization task require different image properties, which also distinguishes our work.**
> >
> > **We also include the new qualitative results in Appendix D of the revised version of manuscript.**
> >
> > **Q4: Table 5, why the False positive sample from [1] is not included?**
> >
> > A4:  Thank you for the suggestion. First, we would like to emphasize that our study and [1] address different application domains. While both involve data synthesis, [1] primarily focuses on vanilla knowledge distillation, whereas our work targets the more challenging scenario of quantization-aware training with lower data precision. **It is noted that in challenging quantization tasks, the synthesized data should be highly fitted to the model statistics in order to calculate the quantization parameters, which makes false positive samples [1] that perform well in the distillation task inadequate for the quantization task.**
> >
> > In our experiments, we utilized synthetic images generated by [1] and applied them to quantization-aware training, maintaining the same fine-tuning strategy as in our approach. The results, shown in Table C, highlight two key observations:
> >
> > 1. False Positive Sampling outperforms baseline methods such as Multi-Sample and Tile (from Table 5), demonstrating its ability to partially replicate the label distribution of the original COCO training set.
> > 2. Our method achieves superior results in most cases, with improvements of 0.6%, 1.0%, 0.7%, and 1.2% at 5-8 bits. Notably, at W8A8, it even outperforms quantization-aware training using real labels, further demonstrating the effectiveness of our sampling approach.
> >
> > **Table C Comparison with False Positive Sampling under the Same Fine-Tuning Strategy (YOLOv5-s on MS-COCO)**
> >
> > | Prec. | Method                      | Real label | Data Distri. | mAP      | mAP50    |
> > | ----- | --------------------------- | ---------- | ------------ | -------- | -------- |
> > | FP32  | Baseline(pre-trained)       | √          | √            | 37.4     | 56.8     |
> > |       |                             |            |              |          |          |
> > |       | Real Label                  | √          | √            | 35.8     | 55.0     |
> > | W8A8  | False Positive Sampling [1] | ×          | ×            | 34.7     | 52.8     |
> > |       | Ours(Adaptive Sampling)     | ×          | ×            | **35.9** | **54.7** |
> > |       |                             |            |              |          |          |
> > |       | Real Label                  | √          | √            | 34.9     | 53.1     |
> > | W7A7  | False Positive Sampling     | ×          | ×            | 33.9     | 52.0     |
> > |       | Ours(Adaptive Sampling)     | ×          | ×            | **34.6** | **53.1** |
> > |       |                             |            |              |          |          |
> > |       | Real Label                  | √          | √            | 32.7     | 51.4     |
> > | W6A6  | False Positive Sampling     | ×          | ×            | 31.0     | 48.2     |
> > |       | Ours(Adaptive Sampling)     | ×          | ×            | **32.0** | **50.0** |
> > |       |                             |            |              |          |          |
> > |       | Real Label                  | √          | √            | 28.0     | 45.8     |
> > | W5A5  | False Positive Sampling     | ×          | ×            | 25.5     | 41.1     |
> > |       | Ours(Adaptive Sampling)     | ×          | ×            | **26.1** | **42.3** |
> > |       |                             |            |              |          |          |
> > |       | Real Label                  | √          | √            | 19.0     | 33.4     |
> > | W4A4  | False Positive Sampling     | ×          | ×            | **16.1** | **27.9** |
> > |       | Ours(Adaptive Sampling)     | ×          | ×            | 15.0     | 27.0     |

---

> > > ### Author Response · Authors · 2024-11-22
> > >
> > > **Q5: What is the purpose of the weak baseline Gaussian in Table 5?**
> > >
> > > A5: Thank you for your insightful question about the purpose of the baseline Gaussian and we would like to provide some clarification here. In Table 5, we use Gaussian noise as input as an ablation study to assess the impact of image quality. For this experiment, we keep the framework's fine-tuning and knowledge distillation setup unchanged, with image quality as the only variable. The resulting QAT-trained network fails to converge, indicating that our adaptive sampling method effectively extracts data characteristics from the pretrained model, and confirming that the effectiveness of our method is not solely due to distillation.
> > >
> > > Additionally, for the Tile and MultiSample baselines, while their label distributions match the original COCO dataset in the in-distribution setting, their object size and location distributions differ from the original data, leading to a drop in accuracy. This result suggests that label, size, and location distributions all play crucial roles in achieving accurate QAT training.
> > >
> > > Reference:
> > >
> > > [1] Akshay Chawla, Hongxu Yin, Pavlo Molchanov, and Jose Alvarez. Data-free knowledge distillation for object detection. In Proceedings of the IEEE/CVF Winter Conference on Applications of Computer Vision, pp. 3289–3298, 2021.
> > >
> > > [2]Xu, Shoukai, et al. "Generative low-bitwidth data free quantization." Computer Vision–ECCV 2020: 16th European Conference, Glasgow, UK, August 23–28, 2020, Proceedings, Part XII 16. Springer International Publishing, 2020.
> > >
> > > [3]Choi, Kanghyun, et al. "Qimera: Data-free quantization with synthetic boundary supporting samples." Advances in Neural Information Processing Systems 34 (2021): 14835-14847.
> > >
> > > [4]Qian, Biao, et al. "Adaptive data-free quantization." Proceedings of the IEEE/CVF Conference on Computer Vision and Pattern Recognition. 2023.
> > >
> > > [5]Chen, Xinrui, et al. "TexQ: zero-shot network quantization with texture feature distribution calibration." Advances in Neural Information Processing Systems 36 (2024).

---

> > > > ### Author Response · Authors · 2024-11-25
> > > > **Gentle reminder: The interactive discussion period will end in less than two days**
> > > >
> > > > Dear Reviewer 3tqE,
> > > >
> > > > Thank you again for your time and efforts in reviewing our paper.
> > > >
> > > > As the discussion period draws close, we kindly remind you that two days remain for further comments or questions. We would appreciate the opportunity to address any additional concerns you may have before the discussion phase ends.
> > > >
> > > > Thank you very much.
> > > >
> > > > Best regards,
> > > >
> > > > Authors

---

### Author Response · Authors · 2024-11-22
**Author Rebuttal by Authors**

Dear reviewers,

We sincerely appreciate the time and effort dedicated to evaluating our work. We have summarized the additional experiments and analyses during the rebuttal phase, and we have included most of them in the revised manuscript.

Our newly added main experiments and analysis in the updated manuscript include:

- **A more granular analysis of the calibration set size across different bit-widths in Appendix B.2:** We conduct an ablation study on calibration set size across a wider range of bit widths, further validating the rationale behind our chosen settings.
- **Add time consumption of generating samples in the first stage:** In Appendix C, we provide additional details regarding the generation of synthetic images.
- **Additional qualitative results of the synthetic data and quantized network:**  In Appendix D, we include additional visualizations of the synthesized data and the quantized network, further demonstrating the effectiveness of our sampling and fine-tuning methods.

We would like to further emphasize our main contributions as follows:

- For the first time, a pipeline for zero-shot quantization-aware training (QAT) for object detection is built, which is verified on both one-stage and two-stage object detection models.
- A novel bounding box and category sampling strategy is proposed to infer the original training data from a pre-trained detection network, allowing for the reconstruction of object locations, sizes, and category distributions without prior knowledge.
- Feature-level alignment is integrated into the QAT process, amplifying its efficacy through the integration of feature-level distillation.
- The proposed method demonstrated state-of-the-art performance in low-bit-width quantization on MS-COCO and Pascal VOC datasets. For example, when quantizing YOLOv5-m to 5-bit, the method achieved a 4.2% improvement in the mAP metric, using significantly less calibration data compared to other methods trained with full train sets.



Please find the point-to-point response with additional details in the following rebuttal section. We sincerely hope that our responses have enhanced the paper's quality and addressed your concerns. If you have any additional suggestions or comments, please don't hesitate to share them. We look forward to engaging in a constructive discussion during the rebuttal phase. Thank you again for your understanding and consideration.

Best regards,

Authors

---

### Meta-Review · Area_Chair_Ys5Y · 2024-12-11

**Metareview:**

The paper proposes a novel zero-shot quantization framework for object detection. Unlike prior ZSQ methods focused on classification, this framework addresses specific challenges in object detection, including class imbalance and localization. The proposed relabel strategy utilizes a pretrained full-precision model to refine bounding boxes and class labels for synthetic data generation and QAT incorporates feature-level and prediction-matching distillation for effective knowledge transfer during low-bit quantization. Extensive experiments validate the effectiveness of the framework on MS-COCO and Pascal VOC datasets, including ablation studies and comparisons with state-of-the-art methods. While the paper addresses an important and underexplored area, the submission has several limitations surrounding presentation clarity and scalability issues as agreed by all reviewers, that collectively fall below the acceptance threshold.

**Additional Comments On Reviewer Discussion:**

- Reviewers (3tqE, R2o8) raised concerns about unclear explanations of the optimization process, the role of losses and inconsistent notations in figures and equations. Authors clarified the roles of losses and updated notations in Figure 2 but acknowledged the computational complexity of the framework.
- Reviewer R2o8 noted the high computational demand for synthetic data generation, questioning its practicality for larger models. Authors highlighted the efficiency of reduced data size (2k) and suggested accelerating data generation as a future research direction.

Ultimately, the paper's novelty and potential impact are recognized, but unresolved concerns about clarity, scalability, and experimental scope led to maintaining the recommendation to reject.

---

### Decision · Program_Chairs · 2025-01-22

Reject